# Effect of Non-Thermal Food Processing Techniques on Selected Packaging Materials

**DOI:** 10.3390/polym14235069

**Published:** 2022-11-22

**Authors:** Domagoj Gabrić, Mia Kurek, Mario Ščetar, Mladen Brnčić, Kata Galić

**Affiliations:** Faculty of Food Technology and Biotechnology, University of Zagreb, 10000 Zagreb, Croatia

**Keywords:** food packaging, biobased, nanomaterials, non-thermal processing

## Abstract

In the last decade both scientific and industrial community focuses on food with the highest nutritional and organoleptic quality, together with appropriate safety. Accordingly, strong efforts have been made in finding appropriate emerging technologies for food processing and packaging. Parallel to this, an enormous effort is also made to decrease the negative impact of synthetic polymers not only on food products (migration issues) but on the entire environment (pollution). The science of packaging is also subjected to changes, resulting in development of novel biomaterials, biodegradable or not, with active, smart, edible and intelligent properties. Combining non-thermal processing with new materials opens completely new interdisciplinary area of interest for both food and material scientists. The aim of this review article is to give an insight in the latest research data about synergies between non-thermal processing technologies and selected packaging materials/concepts.

## 1. Introduction

Tailor-made packaging solutions are used for each and individual food (or beverage) product. Packaging offers convenience in terms of distribution, traceability and effectual marketing. Adequate selection of packaging materials/systems gives as an output a prolonged shelf-life of foodstuffs whilst maintaining the quality and microbiological safety. Due to long processing times and subjection to high temperatures in thermal processing, the quality of food products cannot meet the consumers’ expectations in the 21st century. In that sense, current research is targeted towards in-package and post-package processing in order to achieve adequate decontamination rate of packaged foods. Selection of packaging materials used prior to processing is of high importance due to specific requirements for non-thermal conditions and risk of post-process recontamination [1,2] A recent review papers [3,4] describe progress and new trends in the production of smart, active and intelligent food packaging, with focus on biodegradable and biobased polymers. As for the future research authors emphasized the needs to focus on some important aspects such as development of highly sensitive, eco-friendly and low cost paper-based electrical sensors. Possibilities to scale-up, costs, regulatory aspects, and consumers’ acceptance are also often questionable, making it take a long time for these systems to be commercially available [5]. 

Traditional thermal food treatment results in deterioration of physico-chemical and sensory properties. Nowadays, consumer demands are shifted towards organoleptic superior properties (such as appearance and flavor) as well as in high nutritional value of the product with extended shelf-life. Non-thermal processing technologies (NTP), fulfill the above, so could potentially replace traditional thermal processing methods [6]. They cause inactivation of microbes without the direct application of heat, thus without affecting nutritional values. The most common NTPs are irradiation, ultrasound processing, plasma discharge, high-pressure processing, pulsed electric field, e-beam, and gamma irradiation. In some cases, they can cause the release of free radicals that might interact with packaging material or (inside pre-packed food) with food components, leading to undesirable, and sometimes adverse, reactions. 

In the scope of this review, brief introduction to NTPs will be presented followed by explanations of their impact on properties of biobased polymers, edible films and nanomaterials. Additionally, and if existing in the scientific literature, the evidence of NTPs with real foodstuffs will be presented. A list of abbreviations is given at the end of the manuscript.

## 2. Short Overview of Non-Thermal Food Processing Techniques Commonly Applied on Food Packaging Materials

In the high pressure processing (HPP) method, the pressure is usually transferred by water contained in the high pressure processing chamber. Although this method started to be used more than a century ago, it finally reached the point where it can be applied commercially on a larger industrial scale [7]. The method involves the Le Chatelier and the isostatic principles. It requires relatively low amounts of energy, and can reuse the pressurization fluid (water) with zero waste emission. The compression is caused by the air and the water and alters the anatomy of microbial cell walls. This further weakens the activity of foodborne pathogens without affecting organoleptic properties or nutrient composition [8]. The main disadvantage related to this technique is its impact on the mechanical properties of packaging material. Since it impacts the morphological and thermal properties, it also changes the flexibility of materials and their barrier properties. For in-pack processing, packaging materials must be compressed at least 15% [9], which appears to be limiting factor for use of materials other than petroleum-based plastics. In addition, changes might result in higher migration values. Thus, great attention must be given when using high pressure processing with versatile biopolymers. 

Cold plasma (CP) is a partially or fully ionized gaseous system composed of free electrons, ions, radicals, ultraviolet (UV) photons, and excited or non-excited molecules and atoms. Such an ionized gaseous system can cause the functionalization of the polymer surface, by introducing functional groups, such as hydroxyl, peroxide and ethers, by a cross-linking effect, or by increasing surface roughness (etching) [10,11,12]. It has shown good potential for food decontamination, with limited data about packaging materials. Cold plasma has been shown to affect the quality attributes of food during treatment and storage [13]. For in-package treatment, the product is first sealed inside mostly plastic (or glass) material, with air or with modified atmosphere, and the package is exposed to a strong electric field [14,15]. There are no specific packaging requirements, except the risk of production of free reactive species on the packaging surface must be checked from aspects related to chemical migration. 

Pulsed electric field (PEF) uses electric waves with high voltage amplitude. Short electrical impulses (from microseconds to milliseconds each) of high voltage (typically 10–80 kV/cm) are supplied to the product placed between the electrodes in the chamber. The main advantage of this is the minimal loss of aroma and flavor occurs, with improved physico-chemical and nutritional properties [16]. Economically, although expensive for industrial implementation, PEF technology is cost and time effective. About 50% less electricity is required when compared to low-temperature-low-time thermal treatment [17,18]. Unfortunately, main disadvantages of this processing are: (i) requirement of a high peak power, (ii) limited (rather low) volume of the liquid that can be treated, (iii) inevitable occurrence of electrochemical reactions between the electrode and the food such as electrolysis of water, corrosion and migration. To date, few papers have been published about in-package PEF food processing. Even that there are few special requirements for packaging, high barrier materials used for aseptic packaging seem to be the most appropriate ones, such as ethylene vinyl acetate films filled with carbon-black particles [16,19]. 

Irradiation is a method that uses controlled radiant energy (such as X-ray, electron beams, gamma rays) directed towards foodstuffs in order to destroy microorganisms (such as *Salmonella* spp., *Listeria* spp., *E. coli* O157:H7), parasites and insects.

Electron beam (EB) and gamma irradiation (GI) are common methods used for sterilization of packaging surfaces, phytosanitary treatment, and pathogen elimination. Here, the absorbed dose is defined as one of the most critical factors related to the sterility assurance level. Prior to irradiation processing, prerequisites for food packaging materials are chemical stability (e.g., not prone to depolymerisation), good transparency (allowing the light to pass through to the processed food), in accordance with regulations applied and (desirably) of consumer acceptance [20].

Pulsed light (PL) technology is a method used for the sterilization of foods using very high power and very short duration pulses of light emitted by inert-gas flash lamps [21]. The high-power pulses of radiation can be in the spectra of ultraviolet (UV), visible and infrared (IR) light. The commercialization of pulsed light in-package treated foods requires the development of suitable packaging in order to avoid post-treatment recontamination. For the effective treatment, different aspects should be considered, such as food itself, processing parameters, microorganisms, and packaging material [22].

Ultrasound (US) is defined as a treatment that uses acoustic waves with frequencies between 20 kHz and 100 kHz. Ultrasound treatment is non-ionizing, non-invasive and does not pollute the environment [23]. It has many applications in food industry inducing filtration, crystallization, fogging, emulsification, oxidation and reduction reactions, extraction, drying, softening, sterilization, and cleaning. The activity is due to the phenomenon of cavitation, which includes the formation, growth and collapse of the created bubble. The growth and collapse of the bubbles will result in the formation of high temperatures and pressures in the matrix/cell [24]. However, this non-invasive method could affect the food packaging material if used as an in-pack treatment. Therefore, knowledge about changes in barrier performance, mechanical properties, and the risk of migration of unwanted substances from the material into the food must be seriously considered and studied before applying ultrasound treatment to a food product [25].

## 3. Impact of Non-Thermal Food Processing on Selected Packaging Materials

### 3.1. Biobased Polymers

According to the European Bioplastics Association, biopolymers can be classified as a fossil-fuel based and a biobased material. Both may be biodegradable or non-biodegradable. Biosourced polymers distinguish themselves depending on the source of the raw material used for their production. They can be systemized in three main groups as follows: (a) natural bioproducts (e.g., cellulose, hemicelluloses, starch, chitosan, lignin, pullulan, natural rubbers, plant oils, and fats etc.); (b) microbial produced bioplastics (e.g. bacterial polysaccharides: polyhydroxyalcanoate (PHA) and poly(3-hydroxybutyrate)s (PHB, PHBV, P3HB4HB, P3HBHHX), bacterial cellulose); and (c) chemically synthesized (e.g. poly(lactic acid) (PLA)). A recent market survey showed that more than 20 products are present at the pilot plant level (technology readiness level, TRL, 5), with very few products on industrial scale with TRL between 7 and 8 [26]. Both eco-friendly material alternatives and NTP can be considered to have a low environmental impact, and thus, by combining them an improved sustainability is achieved. Significant impacts of different methods on various biobased polymers are summarized in Figure 1. 

#### 3.1.1. Poly (Lactic Acid) (PLA)

This is compostable and non-toxic biobased aliphatic polyester derived by the fermentation of renewable resources such as starch and/or sugar from corn, cassava, potato or sugarcane. This material has a great mechanical strength and plasticity. It is accepted as GRAS (Generally Recognized as Safe) by the Food and Drug Administration (FDA) and is suitable for use in food and beverage packaging. PLA is commercially used for different shapes production, such as food overwraps, yoghurt cups, lunch boxes, blister packs, water and juice bottles, tableware, kitchen utensils etc. [27]. Further, more elaborated examples are used in active or intelligent packaging materials. Antipack TM stands for PLA/starch-based material incorporated with an antifungal substance which is gradually released to prevent mould growth during the storage period [28]. According to the literature data, the global production volume of PLA was around 190,000 tons in 2019 [29].

Almost a decade ago, [30] gave an interesting review on the effect of high pressure processing (HPP) on the integrity of various polymers including PLA. High pressure processing at 300 MPa during 10 min did not influence the thermomechanical properties of the post-processed films [31]. It was also reported that HPP significantly impacts morphology, structure/crystallinity and roughness if the material is treated with (for example, water) or without food [32]. For processing temperatures above its T_g_ (90–115 °C), it was shown to change amourphous phases to glassy state. This change has a direct and reversible effect decreasing the sorption of the aroma compounds in PLA [33], but at the same time increasing aroma permeation. Moreover, at higher temperatures (90–110 °C), significant opacification takes place, probably due to the re-crystallisation induced by hydrolysis [10,34]. Thus, temperature is considered an important factor for high pressure processing of PLA, which is usually performed at low operating temperatures (below 40 °C). Higher intensities of HPP (above 600 MPa) lead to significant changes in gas permeability, as well as enhanced water resistance of PLA films. However, these values remain significantly elevated compared to commercial plastic based on polyethylene (PE) and/or polypropylene (PP). 

The synergistic effect of HPP and cinnamon oil incorporated in PLA was observed during 3 weeks of refrigerated storage of fresh chicken [31]. *Listeria monocytogenes* and *Salmonella Typhimurium* were efecitevely inactivated. 

Recently, suitability of PLA bottles for HPP apple juice was demonstrated [35]. The treatment of PLA bottles with HPP at 600 MPa for 3 min did not cause alterations in the packaging shape and content, confirming PLA as a valid sustainable alternative to polyethylene terephthalate (PET) bottles for short-term cold storage. This is considered as a significant progress in PLA processing, compared to 10 years ago when PLA materials were shown to be not appropriate due to the unacceptable material embrittlement. 

Thus, PLA seems to be a potential candidate as a packaging material for minimally processed food, with short shelf-life, that require less aggressive processing conditions. 

Atmospheric cold plasma (CP) and dielectric barrier discharge plasmas (DBD) are used to improve the compatibility and hydrophilicity between PLA and thermoplastic starch (TPS) [36], polycaprolactone (PCL) [37] and zein [38]. In addition, plasma treatment can significantly contribute to the overall environmental impact, and hence entail future optimization of processing [36]. Cold plasma was also shown to improve the coating process and absorption capacity of antimicrobial agents on the PLA surface. Functional compounds can in turn be progressively released to packed food to exhibit bactericidal effect [39]. A combination of CP and nisin on PLA decreased the viability of Listeria monocytogenes by 2.25 log CFU/mL, while the control PLA film could not inhibit its growth. 

Ultrasound was thought to accelerate the hydrolysis/transesterification reactions that drive the depolymerisation of PLA. However, from our knowledge no many studies on this subject are available. The effect of ultrasonic on PLA was insignificant compared to bulk erosion/depolymerisation, with appropriate media and salts over the range of treatment [40]. Ultrasonification of PLA with ZnO nanoparticles led to a better morphology, UV shielding, and better antimicrobial activity against *E. coli* and *S. aureus* [41].

Treating pomegranate juice packed in PLA coated PET bottles with PEF (100 L/h flow rate, 35 kV/cm field strength, 281 μs total treatment time) significantly improved the microbiological shelf-life of juices over 84 days [42].

Radiation was shown to cause major structure modifications and changes in the functional properties of PLA [43]. The radiation process leads to the formation of free polymer radicals that can induce two opposite types of reactions: (a) cross-linking (enhanced chemical and thermal properties), and/or (b) chain scission reactions (reduced tensile strength and enhanced hydrolysis rate) [44]. In most cases, one of them is predominant depending on the radiation conditions. A detailed description is well reported in the literature [45]. Radiation treatment makes possible to speed up the composting of PLA packages. EB and GI in doses up to 1.5 MeV and 1200 kGy, respectively, were shown to lead to a higher degradability of PLA. Under radiation, changes in molecular weight, morphology, flexural strength, and degradation profile are induced [46,47,48,49]. For doses around 45 kGy chain scission reactions are predominant over cross-linking, and thus they are defined as critical or transition zone between chain scission and recombination of radicals [50]. Taking into account the ecological impact of degradable materials where degradation process is favourable, in other purposes, such as long term food storage, the degradation should be slowed down. The improvement of thermal stability is not easy to achieve due to the complexity of the degradation mechanism. Treating nanoclay-modified PLA with EB promotes cross-linking and helps in controlling the degradation rate of PLA [44]. Similar effects could be also obtained after modification with compounds with aromatic structure, such as benzene ring [51] and copolymers [52,53], or by increasing treatment temperature above its glass transition [47]. Recently, [53] showed that the presence of SIS (styrene-isoprene-styrene triblock polymers) increased the thermal stability of PLA/SIS blends.

#### 3.1.2. Poly (Butylene-Adipate-Co-Terephthalate) 

Poly (butylene adipate-co-terephthalate) (PBAT) is obtained by poly-condensation between butanediol, adipic acid and purified terephthalic acid. It is considered the most promising aliphatic-aromatic co-polyester [54]. An aliphatic unit in the molecule chain provides good biodegradability, while the aromatic unit provides excellent mechanical properties compared to other biobased polymers. Since pure PBAT still does not satisfy commercial acceptance due to higher costs and mechanical properties not being comparable to synthetic polymers, it is often blended with other biobased polymers to improve its performance.

EB irradiation was used to chemically cross-link poly(glycolic acid)/PBAT blends to impart great barrier properties while maintaining a high toughness of 75 MPa. EB doses below 200 kGy were shown to enhance packaging performance [49,55,56], while doses higher than 300 kGy can cause inhomogeneity on the film surface of the material and degradation of its mechanical performance [57]. 

PBAT was shown to be sensitive to UV radiation, often known as photodegradation [58]. Thus, it is also considered as a perfect candidate for mulch films for agriculture [59]. 

Recently, [60] have shown a synergistic effect between UV-C LED irradiation and antimicrobial PLA/PBAT packaging enriched with grapefruit seed extract and zinc oxide blend: counts of enterohemorrhagic *Escherichia coli* and *Staphylococcus aureus* during storage of fresh-cut onion, cabbage, and carrot were significantly reduced, without using chemical preservative, making it possible to be used as active packaging. 

Ultrasound was shown to contribute to a uniform distribution of PBAT and birch bark (genus Betula) antimicrobial extract [61].

#### 3.1.3. Thermoplastic Starch

Thermoplastic starch (TPS), a material made from starch, water, and/or plasticizers is often considered as the most promising biodegradable material. The major drawbacks of this material are its hygroscopic nature, low gas barrier, and inadequate water barrier. By combining TPS with non-thermal processes, significant changes in material might be observed. 

The electron beam irradiation was shown as the most promising method for sterilization of the TPS/polybutylene succinate (PBS) blend. It provides high efficiency in elimination of microorganisms. The electron beam can induce the cross-linking through the formation of carboxylate and carbonyl groups assigned to the irradiation reaction between the water molecules and the starch. This subsequently leads to the increased moisture resistivity (by up to 10%), higher tensile strength, improved thermal properties and changes in blend degradation [62]. 

Air and oxygen plasma can be efficiently used to modify the surface of cellulose fibers. Consequently, the adhesion between the TPS matrix and modified fiber is better [63,64]. Poly(hydroxybutyrate) (PHB) can be used as reinforcement in thermoplastic corn starch and treating it with sulphur hexafluoride gas (SF6) showed a higher thermal stability but weakened mechanical properties if compared to the untreated composites. Poor mechanical properties were assigned to the agglomeration of PHB particles in the starch matrix that are considered as stress concentrators and thus leading to a weak interfacial adhesion [65]. Ref. [37] used atmospheric cold plasma to improve the adhesivity of PLA and polycaprolactone (PCL) to cassava starch films. Cold plasma led to the increased surface roughness and hydrophilicity and thus improved the adhesion and delamination resistance of produced blends.

Dielectric barrier discharge at atmospheric pressure can be used to assure proper compatibility between PLA and TPS [36]. Plasma treatment can improve the overall environmental impact, but currently there is still need for further process optimization.

#### 3.1.4. Poly(3-Hydroxybutyrate-Co-3-Hydroxyvalerate)

Poly(3-hydroxybutyrate-co-3-hydroxyvalerate) (PHBV) is an aliphatic polyester belonging to the PHA family, and known for its biodegradability, non-toxicity, and biocompatibility. It is similar to the other biobased alternatives, if used alone. PHBV has serious disadvantages when compared to thermoplastics. 

The poor miscibility of PHBV could be enhanced through treatment with gamma irradiation. Likewise, it was successfully blended to PLA [66] and to natural polyisoprene rubber [67]. Resulting materials had better mechanical performance [67]. Since the applied dose of gamma rays is an important factor for changes in material, for doses above 100 kGy it can be considered that the oxidative degradation of neat PHBV and neat PLA caused the main chain scissions that are responsible for a significant decrease in the average molecular weight of materials, and furtherly characteristics of their blends [66]. EB irradiation leads to changes in the chemical structure of neat PHBV, neat PLA and PHBV/PLA blend. For absorbed dose of 10 kGy, the transformation of ester groups to mainly hydroxyl groups might occur [68]. 

#### 3.1.5. Cellulose Acetate

Cellulose acetate (CA) is a derivate of cellulose with excellent optical properties and high toughness. Films can be produced either by melting or solvent-casting technique. In CA films, HPP (200–400 MPa for 5 or 10 min) caused a reduction in tensile strength, Young’s modulus and an increase in elongation at break [69]. Treated films were more luminous and opaque showing also a reduction of water vapor transmission rate and better water resistance. HPP also caused delamination or slight porosity of CA films, and better mechanical stress during heat sealing (250 °C).

It was shown that γ-irradiation (cobalt-60 source up to 50 kGy) of cellulose acetate cause a decrease in the values of relative, specific, reduced and intrinsic viscosities, molar mass, hydrodynamic volume, real and ideal chain dimensions [70]. Irradiation treatment (10 kGy) also improved the quality of nanocomposite films (cellulose acetate-polyethylene glycol/clay) (reduced opacity index, improved mechanical properties, and water and oxygen permeability) [71]. An interesting book chapter covers issues related to irradiated cellulose for use in food and agriculture, with the insight in the enlarged application of irradiation-induced biodegradable polymers instead of nonbiodegradable ones [72]. 

#### 3.1.6. Polyhydroxyalkanoates

Polyhydroxyalkanoates (PHA) are linear polyesters, insoluble in water that naturally occur in a variety of microorganisms. They are produced by the fermentation of renewable carbon resources and are accumulated intracellularly as energy reserves. They exhibit thermoplastic properties. The most commonly used PHAs are polyhydroxy-butyrate (PHB, C4) with a methyl group as a side chain, or poly-3-hydroxybutyrate-co-3-hydroxyvalerate (PHBV, C4 resp. C5) which is a copolymer of PHB and PHV with a methyl or ethyl side chain, respectively. Due to the longer side chain of PHV in comparison to PHB, the melting and the glass transmission temperatures are lower. This leads to a lower crystallinity and a better processing behavior of the extremely thermosensitive PHB-(co)-polymers in an extrusion process. 

Picosecond pulsed laser treatment, compared to UV irradiation [73], showed its convenience for PHAs micro structuration in order to improve the biocompatibility for scaffold manufacturing in the food packaging and tissue engineering fields.

To our knowledge, there are no other scientific articles published combining PHA with non-thermal food processing.

#### 3.1.7. Edible Coatings

The use of edible coating in combination with some non–thermal processing became a trend in the last decade. There are excellent review articles on edible film/non-thermal processing/nanomaterials [74], edible film/pulsed light [22,75], and edible coating/ozonation/gamma irradiation [76]. The prevalence of scientific articles on this subject published in science direct, only in the last 2 years is given in Figure 2. Therefore, in this article only some examples not already covered in the literature will be mentioned in order to review of possible effects on both food and applied coating material. 

Gamma irradiation, in combination with EC, has been a well-studied topic in recent two decades [77,78]. Doses below 2 kGy are shown to retard sprouting of vegetables and fruits’ ripening, medium doses (from 1 to 10 kGy) to reduce pathogens (similar effect compared to conventional pasteurization), while doses above 10 kGy are required to reach a sterile food product. Irradiation can be used to enhance cross-linking in protein-based coatings (covalent protein molecules link) [24,79,80]. Irradiation in doses from 5 to 20 kGy was shown to extend the shelf-life of Hayani date fruit (*Phoenix dactylifera* L., Family *Arecaceae*) coated with poly(vinyl alcohol), PVOH, chitosan and tannic acid from 7 to 30 days [81]. Irradiation doses of 0.25 and 1.0 kGy with commercially available carnauba wax coating improved the shelf-life of tamarillo (*Cyphomandra betacea*) and Jujube fruit (angiosperm clade) [1]. In another study, combining chitosan coating and UV radiation (253.7 nm for 6 min) helped in controlling the respiration rate of jujube fruit (*Ziziphus jujuba*) whilst preserving ascorbic acid and chlorophyll [82]. Irradiation of strawberries was shown to inhibit *Botrytis cinerea* and other visible fungus and molds in combination with various coatings: protein/limonene/peppermint coatings (irradiation at doses of 32 kGy) [83], with milk protein/*Quillaja saponaria* EC (irradiation at doses of 15-35 kGy) [84] and carboxymethyl cellulose (irradiation at doses of 2 kGy) [85]. An increase in shelf-life from 9 to 19 days was also reported for banana fruit when treated with 5.87 kGy/h and combined with PVOH/carboxymethyl cellulose/tannin edible coating [86]. 

With regards to vegetables, carrots, tomatoes and broccoli were successfully treated whilst using EC prior to irradiation. An interesting two-step post-treatment processing of carrots showed improvement in firmness, sample weight, and color [78]. Irradiation at 32 kGy was used on pre-cut carrots coated with calcium caseinate and then samples were treated with nanoemulsion based coating enriched with citrus extract, cranberry juice and essential oils. Irradiation also led to a better mechanical properties and a water vapor permeability of used coatings. Same authors found that combining active EC (based on thymol, winter savory, Ceylon cinnamon and lemongrass) and irradiation at doses of 0.4 and 0.8 kGy kill pathogens effectively (*E. coli* O157:H7, *L. monocytogenes*, *S. Typhimurium* and *A. niger*), resulting in extended shelf-life of ready-to-eat broccoli [81]. 

Gamma irradiation is also known as an efficient method for meat and fish decontamination. For example, when used in combination with thyme/cannelle/oregano, the shelf-life of chicken might increase by 14 days [87]. 

In the past decade, pulsed light in combination with edible coatings was shown as effective in industrial proof-of-principle trials for fruits and vegetables [75]. “Golden Delicious” apples were treated with pulsed light (spectrum range of 180–1100 nm; 30 pulses; 0.3 ms, total energy accumulated 12 J/cm^2^) in combination with gellan gum [88] or pectin based coating [89]. Great synergistic effect resulted in reduced tissue softening and browning whilst preserving antioxidant value of apples, reducing microbial loads and without affecting freshness of apples. Similarly, [90] demonstrated an extended shelf-life of mango fruit when treated with pulsed light (8 J/cm^2^) and alginate coatings. Preservation of the antioxidant capacity, color, and reduced *L. innocua* loads make this method a promising tool for commercialization of such high nutritive fruit for countries which do not produce mango themselves [90]. Pulsed light with pectin, alginate or gellan was also used for extending the shelf-life of fresh-cut cantaloupe melon up to 28 days [91,92,93]. Treatments with chitosan resulted only in improved microbiological quality but negatively influenced ascorbic acid content and loss of fluids [92], while alginate maintained physico-chemical and nutritional quality and reduced fluid loss [93]. 

A combination of chitosan (1%)/mandarin essential oil (0.05%) and pulsed light (3, 6 and 12 J/cm^2^) on fresh-cut cucumbers and fresh green beans had no effect against *L. innocua*, but coating maintained tissue integrity and sample firmness [94]. 

Recently, in an interesting research study, the authors showed that PL of 1.32 J/cm^2^ did not affect albumen quality with maximum of 3.77 log CFU/egg of *E. coli* inactivation and thus possibility of using pulsed light with Vaseline coating on whole shell eggs for industrial application [95]. 

Plasma treatment (V = 60 and 70 kV, 1–5 min) of sodium caseinate results in an improvement in the hydrophilic properties, film roughness and surface oxygen content due to the increase of O−H and C=O groups [96].

Ref. [97] used ultrasonic pretreatment (19 W, 5 min at room temperature) to improve the interaction between kidney bean protein isolate and chitosan to produce films with increased opacity.

Some of the results pertaining to studies on the non-thermal effects on edible packaging are shown in Table 1.

### 3.2. Nanomaterials

Recent developments in nanotechnology have enabled the synthesis of nanoparticles with high stability that can be used directly in food or in food packaging materials. They are often applied to tailor-made design to improve properties of packed product. Novel process and types of nanoparticles allow reductions in degradation and cause the inactivation of incorporated nanoparticles after their application. In packaging field, nanotechnology has been used to improve the material performance (especially its barrier and thermal properties) or in order to incorporate various active compounds, such as antibiotics, antimicrobials or antioxidants [74,104,105]. Nano-materials (nanoparticles, nanoemulsions, nanocomposites, nanostructured materials) can be used as (a) new active functional packaging; (b) as new smart packaging materials such as (bio)sensing technologies for detection of nutritional and non-nutritional components, antioxidants, adulterants and toxicants; and (c) as enhancers of barrier and mechanical properties the existing food packaging materials [74,104,106,107,108,109,110,111,112,113,114,115]. In addition, they are used to extend food shelf-life, while reducing waste and ensuring adequate food safety and quality. The environmental, health and safety implications of nanomaterials in the food sector are important issue taking into account regulation and consumer perception of these materials [114].

Nanoscale fillers, such as clay and silicate nanoplatelets, silica (SiO_2_) nanoparticles, chitin or chitosan, can be applied into the polymer matrix to provide lighter, stronger, fire resistant, and better thermal properties of materials [106,116]. Antimicrobial nanocomposite films, made by impregnating the fillers into the polymers offer desirable barrier properties (to moisture, water vapor, gases, and solutes) for specific food products [117,118]. To our knowledge, there has been little research performed on the effect of NTP on the nano-packaging materials used for in-package food processes. Thus, in the context and the scope of this article, the focus will be done on those nanoparticles in biomaterials intended for use in combination with NTP techniques. 

Some of the interesting examples found in the literature [94,119,120,121] will be presented in the following paragraphs (Table 2).

Although biobased materials have many advantages compared to petroleum-based resins, they often lack barrier properties. Combining HPP with biopolymers often results in a decrease of their barrier performance. In this concern metal nanoparticles (such as zinc, silver, or titanium) and nanofillers from graphene family (such as graphene sheets, graphene oxide, and expanded graphite) are shown to be promising enhancers of gas and water vapor barrier, thermal stability, and tailored functionality. Silver nanoparticles (nano-Ag) have been used in food packaging applications due to the remarkable antibacterial properties and high thermal stability. Since HPP can cause changes to the morphology of the polymeric materials, the mass transfer of permeants in the material or from the food to the material should be considered [123,124]. The addition of nano-Ag could impact the microstructure, and thus improve the thermal, mechanical and barrier properties of the PLA nanocomposite film treated with HPP [125]. Moreover, nano-Ag has a strong inhibitory effect on various pathogenic microorganisms such as *E.coli*, *Neisseria gonorrhoeae*, and *Chlamydia trachomatis* [126]. 

Graphene based nanoparticles have a unique conductivity, single layer formations, and strong reinforcement capability when compared to the other nanocomposites. Therefore, graphene oxide has been the most versatile and the most commonly used in production of composites and thin films for improving the gas barrier [127]. For example, in PLA, its presence leads to a significant morphological change (increased roughness) and decreased tensile strength, while if treated with HPP (300−600 MPa/15 min), glass transition and crystallization temperatures are increased indicating improvements in thermal stability. The loading concentration of graphene oxide and applied pressure are shown to play an important role in the changes caused by HPP. In other words, the barrier properties of PLA were first modified due to the presence of graphene oxides and later by the pressurization. The authors demonstrate major limitations of biodegradable single-layer composite films in a high pressure environment leaving their use in industrial purposes still questionable [32]. 

HPP (especially at 600 MPa) in combination with nano-TiO_2_ improved the mechanical (elongation-at-break) and the barrier (water vapor and oxygen) properties of the biodegradable polyvinyl alcohol (PVOH) and chitosan composite films [128]. The antibacterial activity of films was also enhanced. The HPP treated PVOH–Chitosan–TiO_2_ films also showed great stability in food simulants (distilled water, acetic acid, ethanol and olive oil) which makes this composite material adequate for food packaging. Cotton linen nanowhiskers processed in a high-pressure homogenizer and used as a reinforcing material in thermoplastic starch improved performance of composite material [129]. 

Electrically conductive polymers can be used to provide the necessary electrical conductivity needed for electrically employed treatments, such as PEF. The main disadvantage for use in food applications is a lack of biocompatibility and biodegradability and often a cytotoxic character. Recently, there has been research interest in novel non-toxic electrically conductive nanofillers made from/or in combination with natural biopolymers, such as those with graphene oxides, naturally occurring clays, or multi-walled carbon nanotubes [130]. The food packaging materials for in-pack PEF treatment should have a through-plane electrical conductivity close to the electrical conductivity of the packaged food, which is typically between 0.1–2 S/m [131]. Since many biopolymers have only in-plane conductivity, their application for in-package PEF food treatments remains subject to future investigations [132]. The combination of bioactive nanoparticles with PEF opens new opportunities in overcoming the resistance of materials and to improve inactivation efficacy without noticeable losses of food quality [132]. Ref. [133] found that the application of microsecond-range PEF in combination with the encapsulated nisin nanoparticles can significantly potentiate the efficacy of the antimicrobial treatment even against bacteria in stationary growth phase. Even though this study promises great results, authors underlined that there is still a lack of sufficient electrophoresis. Thus, it remains the major limitation of nanosecond PEF protocols in food industry where mass transfer is required.

Recently, Refs. [130,131] developed electrically conductive bionanocomposite using chitosan and reduced graphene oxide. The addition of biocomposites resulted in an increased mechanical and water resistance, in addition to the electrical properties, making this material suitable not only for use in food packaging but also other sectors (electric field processing, body sensors and electro-responsive biocompatible devices). Graphene nanoparticles can be hardly used in pure PLA since it has a weak affinity for graphene. Thus, blending it with another biodegradable polymer with a good affinity for graphene nanoparticles, like PBAT, seems a good strategy to hold high loadings of this conductive filler. Recently it was shown that confining nanofillers into the PBAT continuous phase resulted in the creation of conductive channels and good percolation networks, with an electrical conductivity of 338 S/m. Multi-walled carbon nanotubes were also used in small amounts as doping materials to provide the electrical conductivity and desirable mechanical properties to chitosan [134]. When the content of the nano-sized carbon fillers is increased, then light scattering inside the matrix becomes more probable, so transmitted light intensity decreases. The authors of [134] indicated that if the target is to increase the electrical conductivity that might be achieved with high loadings, then it is sometimes necessary to sacrifice optical transparency. 

Electron beam has been recently used to improve the interfacial compatibility between polar nanofillers (graphene based or natural clays) and non-polar polymer resins such as PLA. The main principle is in removing the surface impurities and altering the surface chemical characteristics under the proper irradiation conditions. Halloysite nanotubes, which are clay particles with a tubular shape present in nature, have been recently used as reinforcing fillers. For example, after electron beam exposure, halloysite-enriched PLA samples showed increased crystallinity and consequently higher T_g_ value (from 310 to 316 °C of non-irradiated and irradiated samples, respectively). Additionally, at lower irradiation doses the lower water vapor permeability was measured (decrease of 20% for 20 kGy compared to 40 kGy irradiation) [135]. PLA can also be modified with the montmorillonite (MMT). An increase in the mechanical properties and oxygen barrier compared to the neat PLA was reported. This study also demonstrated that PLA nanocomposite films are suitable materials for the irradiation processing of prepacked food at doses from 1 to 10 kGy [44].

The addition of bio-CaCO_3_ nanoparticle in the electron-beam irradiated (100 and 200 KGy, using a 1.5 MeV electrostatic accelerator) PBAT/starch and PBAT/PLA blend improved the thermal resistance of material [136,137]. The incorporation of organomodified montmorillonite (OMMT) contributed to the enhanced miscibility, thermal stability of PHBV, PLA, and PHBV/PLA blends that is usually observed only when the clay is homogeneously dispersed. The gamma irradiation of PHBV/PLA/ OMMT with the compatibilizer lead to the oxidation reactions of ester groups, inducing the formation of hydroxyl groups, and thus improving the miscibility. Gamma irradiation at doses of 10-40 kGy was used to improve the dispersion of chitin nanowhiskers (1–5 wt %) in PCL films. Irradiation led to the reduction of chitin macroscopic domains when compared to the production without the irradiation, but no significant effect was observed depending on the irradiation dose. Additionally, irradiated materials had better mechanical properties, as well as reduced yellow discoloration of chitin by irradiation. 

UV radiation can be used to accelerate the degradation of PBAT films modified with organoclay Cloiste C20A at concentration up to 5% [58]. It was also reported that the chitosan-TiO_2_ nano film showed a decreased transmittance in the visible light region, so UV light facilitated its photocatalytic antimicrobial effect under ambient condition [1].

The effect of cold plasma treatment (Table 2) was used to improve surface morphology, barrier properties, and the release efficiency of active substances from nanofibers [121,122]. Ref. [138] showed that the ratios of oleic acid to stearic acid and ultrasonic/microwave assisted treatment had a significant effect on the WVP and on the contact angle of prepared films. It was found that the oleic acid: stearic acid = 2:3 films had the lowest WVP value (0.1 × 10^−12^ g/cm s Pa) and the highest contact angle value (135°), at treatment conditions of 20 °C, 15 min, and 500 W. 

The ultrasound was used for the incorporation of starch nanoparticles in PBAT/TPS blends without the addition of any chemical reagent. The ultrasound treatment resulted in the formation of starch less than 100 nm in size and of an amorphous character, lower thermal stability and low gelatinization temperature when compared with cassava starch films [139].

## 4. Active Packaging

If the packed food is treated with one of the non-thermal techniques, the addition of the active compound should not affect the active properties of the whole package nor its primary preservation role. Thus, the final efficiency combines the synergistic effect of both non thermal principles. In recent years, the combined use of essential oils (EOs) with NTP has attracted much attention for microbial inactivation. It is due to the fact that EOs have often been proposed as food preservatives due to their strong, wide-spectrum activity against microorganisms. However, because of the strong organoleptic impact they cannot often be used in excreted doses. In the combination with other inactivation mechanism a good efficiency might be achieved even after the application of a lower dose of the EO. An interesting review article was recently published covering the issue [140].

If the target function is the antimicrobial effect, then the NTP will be used to lower the initial microflora counts and to inhibit their enzymatic activity, while the active compounds will limit further microbial growth. The reduced growth can be achieved either by controlled release of the antimicrobial substances or by the degradation of product during the storage by controlling the gas composition and moisture in the package space. 

In addition to the antimicrobial function, the NTP can also cause the oxidation of food components. For example, in protein rich food, HPP leads to the intracellular release of prooxidants and starts the catalytic oxidation reactions [141]. Then, combining novel techniques with active antioxidant packaging can be beneficial. The most recent literature examples are given in Table 3.

Irradiation (ultraviolet, gamma, electron beam, ion beam, and laser exposure [152]) is a method used in combination with active packaging (AP) in order to tailor the release behavior of the active agents from the AP systems. The major mechanism of the irradiation action on controlling the release is its ability to induce the cross-linking of functional groups of active compounds to polymer surfaces [153]. Ref. [100] found that the electron beam irradiation (60 kGy) could favor the interactions between antioxidants and the biopolymer via a free radical-mediated mechanism providing a better protection and less mobility within the matrix structure. Irradiation induced cross-linking of biobased polymers can thus reduce the effective diffusion of natural phenolics from the matrix [147,154]. 

For prepacked food products, an important issue is that during the irradiation treatment, the formation of free radicals in food might occur. Moreover, in food with high water content, the reaction between oxygen molecules dissolved in water with hydrogen radicals takes place resulting in the formation of free radicals that in a second step leads to their reaction with small amounts of the remaining oxygen in the packaging headspace. In order to reduce the impact of this process-induced oxidation, combining active O_2_ scavengers with IR can act synergistically. 

## 5. Safety Issues

Safety of food contact materials (FCMs) and articles depends on containing substances which can migrate into food under conditions of intended use [155,156,157]. Moreover, the purpose/type of packaging must be taken into consideration especially if the active packaging is within the context. It is important to clearly distinguish the undesired interactions between the package and the food (migration of the unwanted substances) from those expected and desired, such as in active packaging. The EU Regulation 450/2009 describes that for the active packaging, the migration of active substances must not be taken in the calculation of the overall migration. Still, NTP can influence the migration of the active film ingredients. In other words, with the active packaging developments, a migration of active component becomes a desirable process. Some examples include the migration of the antioxidant agents (such as EOs) and hydro-alcoholic extracts incorporated into chitosan films. Desirable interaction between phenolic compounds from hydro-alcoholic extracts and chitosan can protect the packaged food by acting as an improved barrier against light, water, and oxygen from the outer environment, etc. Additionally, biopolymers with EOs showed exponential diffusion growth, and the active compound present in the simulant kept its antioxidant activity. EOs showed a faster release toward fatty food simulant, highlighting their potential use for fatty food [158]. 

As it was shown, various NTPs can lead to various changes in biobased materials, so it is also questionable if those structural changes can promote migration of additives and other residual compounds that might become food contaminants when released into foods that are in direct contact with materials. For example, irradiation (e.g., γ-rays, X-rays, accelerated electrons, and ion beams) may lead to disproportion, hydrogen abstraction, arrangements, degradation, and/or the formation of new bonds in polymer network [159]. Moreover, pure biopolymers are generally less stable and present a worse water vapor barrier compared to the conventional polymers, more additives are usually used for their production. Therefore, some undesirable interactions and consequent migration of substances may occur [160]. Nonetheless, and to the best of our knowledge, there are substantial gaps in the scientific information available dealing with the effect of NTP on the interactions on the interphase between food and packaging.

Safety concerns associated with the migration of nanoparticles from the packaging material into the food and their impact on the consumer’s health have been in the focus of many papers [161,162,163,164,165,166]. Since 2006, the European Food Safety Authority (EFSA) intensively follows the developments in nanotechnology including reviewing the current state of knowledge and latest developments in nanotechnology regarding food and feed. Information from the European Commission on the regulatory aspects of nanomaterials are published in 2008 [167]. So far, this is an ongoing issue, and additional studies must be performed to examine the risk of nanoparticles due to their properties and capacity for absorption and migration and the effect on human health [168,169].

According to the studies performed, there are many factors that should be taken into account when performing such research (such as particle dissolution, surface morphology of the particles, concentration, surface energy, aggregation, and adsorption).

The effects of the NTP on the food-packaging interactions, i.e., on the chemical migration, are the focus of review paper [170]. More than ten years ago, authors asserted that only a few publications were found on this issue related to the active and the biodegradable packaging systems. The situation has not changed a lot since then, and this observation is valid for the nanomaterials as well. Nanomaterials, as any other material, may have a diverse set of properties that need to be determined and taken into account [171]. However, the few papers published so far are sometimes contradictory with a lot of controversy about the toxicity of nanoclays [172,173].

For the packaging materials used in irradiation process, it is required that they should not transmit any radiolysis product to foods. Formation of radiolysis product (such as low molecular weight aldehydes, and acids) depends on the absorbed dose, atmosphere, temperature, time after the irradiation, and food simulant used [174]. Due to the possible occurrence of radiolysis product resulting from the scissioning or the cross-linking of polymers, as well as from the reactions of additives (antioxidants, stabilizers, etc.), their migration must be evaluated in any pre-market safety assessment prior to their commercial use [175]. 

UV irradiation is one of the primary causes of polymer degradation. It can lead to the formation of oligomers and monomers. Since they are detached from the structured polymer network, then they can migrate into food and influence the biodegradation. The authors of [176] found that no unwanted substances migrated from PLA and cellulose based materials into aqueous food simulants aimed for packaging of fresh-cut cherry tomatoes after UV exposure. 

The trace amounts of TiO_2_ from the PVOH/Chitosan–TiO_2_ film was detected in an olive oil food simulant at 23 °C after the HPP treatment (200–400 MPa), while no other detectable substances were found in distilled water, acetic acid and ethanol [128]. HPP treatment significantly reduced the migration of TiO_2_ nanoparticles from the films, probably by decreasing the molecular chain mobility of PLA while tightening the network structure in nanocomposites (better stiffness and tortuosity) [126]. In this context, PLA was also shown as suitable for HPP due to the low scalping of aroma compounds from food [35]. 

The migration of various packaging additives (antioxidant, Irganox 1076, and ultraviolet light absorber, Uvitex OB) from the commercial PLA, during and after two high-pressure/temperature treatments was measured. The effects of HPP pasteurization (800 MPa for 5 min, from 20 to 40 °C) and HPP sterilization (800 MPa for 5 min, from 90 to 115 °C) were compared with the conventional pasteurization (30 min at 0.1 MPa and 63 °C) and sterilization (20 min at 121 °C) treatments. Migration to the food simulants (distilled water, 3% acetic acid, 15% ethanol and olive oil) was evaluated. The migration of Uvitex OB was very low or no detectable for all the cases studied and comparable to commercial LLDPE [177].

After the 90-days of storage, the migration of nano-Ag particles (AgNPs) from the PLA/AgNPs composite film treated under 200 MPa was low, while it was accelerated under the 400 MPa HPP [178]. 

Chemical migrations have not been observed when using biopolymer-based food packaging, such as PLA films treated with cold plasma [11].

## 6. Conclusions

In the last decade the scientific community is facing an intensive research in emerging food processing technologies looking for products with excellent quality and adequate safety. Parallel to this, an enormous effort is also made in the field of food packaging, aiming to decrease the negative effects of synthetic polymers not only on food product but on the entire environment as well. However, the introduction of new and different packaging materials is under debate, and requires time for their safe application to specific food products. Knowing the number of combinations of existing materials, as well as searching for environmentally friendly ones, implied that this is a never-ending task. Furthermore, the effect of the processing treatments on both food product and the packaging material should be simultaneously investigated in order to result in safe food products on the market. Thus, the selection of the non-thermal food treatment and the optimal eco-friendly packaging can result with products that are acceptable not only to consumers (due to their safety and high nutritional and sensorial attributes) but also to the environment.

## Figures and Tables

**Figure 1 polymers-14-05069-f001:**
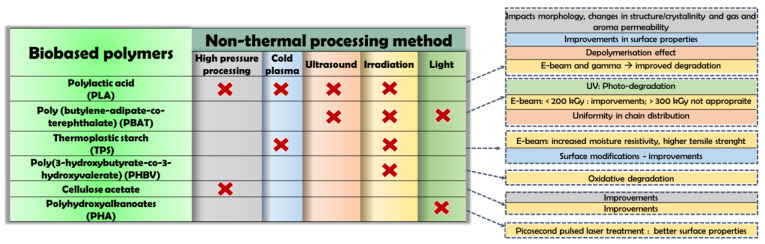
Principle impact of non-thermal food processing on various biobased polymers.

**Figure 2 polymers-14-05069-f002:**
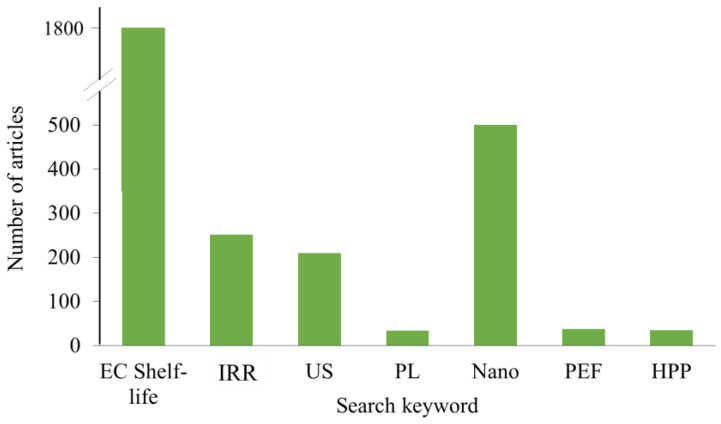
Number of scientific articles found in Science direct in the period from 2017 to 2022. EC—edible coating, IRR—irradiation, US—ultrasound, PL—pulsed light, Nano—nanomaterials, PEF—pulsed electric field, HPP—high pressure processing.

**Table 1 polymers-14-05069-t001:** Effect of non-thermal treatments on characteristics of edible packaging materials.

Packaging Material	Treatment	Effect of Process on Packaging Mterial	Reference
κ-carrageenan/starch blend film	HPP 14 MPa (2–5 passes) and 20 MPa (2 passes)	Increased water resistance and WVP; increased surface hydrophobicity and tensile strength	[98]
Gelatin-based films	HPP (600 MPa), 30 min at 20.5 °C	Decrease in OTR; significant increase in TS and T_m_; Significant decrease in WVTR	[99]
Calcium caseinate- whey protein isolate-glycerol film	γ-Irradiation of 32 kGy	Increased puncture strength; no detrimental effect on WVP	[83]
Chitosan-gelatin films +5 wt % quercetin	Electron beam irradiation of 60 kGy	Decreased the release rate of quercetin; 42% increase in TS; 65% decrease in O_2_P; improvement of thermal stability	[100]
Chitosan (1.5%) coating on fresh jujube fruit	UV-irradiation 253.7 nm; 4 and 6 min	Reduced decay of jujube fruit	[82]
Starch-based film	HMDSO cold plasma, 70 w, 30 min	Increased film crystallinity; improved WVP and mechanical properties of films	[101,102]
Chitosan-based + zein coatings	Cold plasma 100 W (65 V, 1.5 A), d = 5 mm, 30 s	Improved surface wettability; slower drug release rate within 24 h from 72.8% to 49.3%	[38]
Gluten-based film	Ultrasound 600 W/cm^2^, 24 Hz, 3–12 min	Enhanced protein dispersion and the appearance of film	[103]
κ-carrageenan/starch blend film	HPP 14 MPa (2–5 passes) and 20 MPa (2 passes)	Increased water resistance and WVP; Increased surface hydrophobicity and tensile strength	[98]
Gelatin-based films	HPP (600 MPa), 30 min at 20.5 °C	Decrease in OTR; Significant increase in TS and T_m_; Significant decrease in WVTR	[99]
Calcium caseinate- whey protein isolate-glycerol film	γ-Irradiation of 32 kGy	Increased puncture strength; no detrimental effect on WVP	[83]
Chitosan-gelatin films +5 wt % quercetin	Electron beam irradiation of 60 kGy	Decreased the release rate of quercetin;42% increase in TS; 65% decrease in O_2_P; improvement of thermal stability	[100]

HMDSO—Hexamethyldisiloxane; HPP—high pressure processing; OTR—oxygen transmission rate; O_2_P—oxygen permeability; WVP—water vapour permeability; WVTR—Water vapor transmission rate; TS—tensile strength; EAB—elongation at break; T_m_—melting temperature.

**Table 2 polymers-14-05069-t002:** Effect of non-thermal treatments on nano-packaging material characteristics.

Packaging Material	Treatment	Effect of Process on Packaging Material	Reference
PA/LDPE PA/nano/LDPE PA/EVOH/LDPE	Pasteurization 75 °C, 30 min	OTR > 13.3%; WVTR > 96.7% OTR > 75.9%; WVTR > 40.7% OTR < 44.5%; WVTR > 43.8%	[120]
PA/LDPE PA/nano/LDPE PA/EVOH/LDPE	HPP 70 °C, 800 MPa, 10 min	OTR > 16.9%; WVTR > 21% OTR > 39.7 %; WVTR > 21.2% OTR < 53.9%; WVTR > 48.9%	
PA/PP PA/nano/PP	121 °C, 30 min	OTR > 63.3% OTR > 112.6%	
High and low molecular weight (M_W_) PA6 and PA66 silica nanocomposites; Commercial nanocomposites	Temperatures from 20 to 120 °C	Yield stress increases with the addition of layered silicate; Low M_W_ PA6 and PA66 nanocomposites show very brittle fracture behaviour at room temperature; High M_W_ PA6 nanocomposites are ductile; Commercial nanocomposites are brittle; With temperature increase all the nanocomposites become ductile at a certain temperature	[119]
Bioactive coating: 3% N-palmitoyl chitosan + mandarin EOs nanoemulsion	HPP 200–400 MPa, 25 °C, 5 min; pulsed light 3 × 10^4^– 1.2 × 10^5^ J/m^2^	HPP caused disintegration of the coating layer; pulsed light treatment did not affect samples firmness during storage, nor coating integrity	[94]
Thyme EOs/silk fibroin nanofibers	Cold plasma 400W, 4 min; N_2_ flow rate = 100 cm^3^/min	With silk fibroin increased, from 50% to 100%, moisture content increased from 11.87% to 15.77%; water solubility increased from 52.54% to 63.54%; WVP decreased from 1.58 to 0.77 g mm/m^2^ h kPa; TS decreased from 12.9 to 6.53 MPa; EAB increased from 17.06 to 21.39	[122]
Phlorotannin (PT) encapsulated in *Momordica charantia* polysaccharide (MCP) nanofibers	cold plasma 30 s, 350 W, N_2_ flow rate = 100 cm^3^/min	Release efficiency of PT from the nanofibers was enhanced by 23.5% (4 °C) and 25% (25 °C); Antibacterial and anti-oxidant activities of PT/MCP nanofibers were markedly improved; moisture content and water solubility of the MCP nanofibers increased (from 4.28% and 10.42% to 8.91% and 18.94%, respectively); maximum TS was achieved when MCP:PT was 6:1; free radical scavenging capacity of PT/MCP increased to 91.74%	[121]

OTR—oxygen transmission rate; WVTR—water vapor transmission rate; TS—tensile strength; EAB-elongation at break; HPP—high pressure processing; PA—polyamide, EVOH—Ethylene vinyl alcohol; MW—molecular weight; EOs—essential oils; PT—Phlorotannin; MCP—*Momordica charantia* polysaccharide; >increased property; <decreased property.

**Table 3 polymers-14-05069-t003:** Effect of non-thermal treatments on active packaging material characteristics.

NTT	Conditions	Active Substance	Active Character	Food	Effect	Reference
HPP	800 MPa, 10 min. at 5 °C	Rosemary extract 0.45 mg/cm^2^ on LDPE	AO	Chicken patties	HPP reduced the microbial growth and the rosemary suppressed the lipid oxidation	[142]
	600 MPa, 7 min, water at 10 °C	Rice bran extract on internal surface of vacuum package film	AM, AO	Dry-cured Iberian ham	HPP+AP does not improve activity of AP film	[143]
	600 MPa, 8 min	Chitosan, nisin and phytochemicals from rice bran	AM	Sliced dry-cured Iberian ham	HPP+nisin or oryzanol chitosan based-films reduced the population of *L. monocytogenes* by 6 log CFU/g	[144]
	600 MPa, 7 min	Olive leaf extract on internal surface of vacuum package film	AO, AM	Sliced dry-cured shoulders	AP not efficient to preserve the volatile compounds profile of the samples from the changes induced by HPP	[145]
	500 MPa, 2 min at 20 °C	Oregano EOs +Na-alginate edible film	AM	Sliced ham	Reduction of *Listeria* counts below the detection limit	[146]
EBI	60 kGy	Ferulic acid and tyrosol incorporated into chitosan–gelatin edible films	AO	Food simulant (water) at 25 °C	Effective diffusivity of tyrosol was 40 times greater than that of ferulic acid.	[147]
	40 and 60 kGy	Quercetin incorporated into chitosan-gelatin edible film	AO	Ethanol 30% (*v*/*v*) at 25 °C	Irradiation induced a reduction of the quercetin release rate. Effective diffusion coefficient of quercetin was not significantly modified by the irradiation.	[100]
OZ or γ irr	OZ = 10 ppm, 15 min; γ irr = 1 kGy	Alginate/EOs + citrus extract	AM	*Merluccius* sp. fillet	Increased shelf-life of fish fillets from 7 days (control) to 28 days for alginate/EOs/γ irr samples; and 21 days for alginate/EOs/OZ treatment	[148]
γ irr	Low dose γ irr	EOs: Thyme + Cannelle + Oregano	AM	Boneless chicken thigh samples	Shelf-life of the chicken sample increased by 3 days and 8 days when treated with Thyme + Cannelle + Oregano EOs and gamma irradiation, respectively. γ irr + EOs increased shelf-life by 14 days	[87]
	2 kGy	Pectin + curcumin NPs + ajowan EOs nanoemulsion	AM	Chilled lamb loins	Increased shelf-life of lamb loins from 5 days (control) to 25 days	[149]
	1 kGy	Chitosan (film + EOs; Chitosan + Silver NPs (AgNPs); Chitosan + Eos + AgNPs	AM	Strawberry	Strong AM activity against *Escherichia coli, Listeria monocytogenes, Salmonella Typhimurium*, and *Aspergillus niger*. All composite films exhibited lower weight loss than control samples, and γ-irr reduce the firmness and decay during 12 days of storage	[150]
	2.5 kGy	Chitosan + Cumin EO nanoemulsion	AM	Beef loins	Effective to control microbial population; Enhanced storage life (~ 14 days) of beef loins and slowed some physico-chemical changes	[151]

AP—active packaging; LDPE—low density polyethylene; EOs—essential oils; AO—antioxidant; AM—antimicrobial; γ irr—gamma irradiation; OZ—Ozonation; HPP—high pressure processing; EBI—electron beam irradiation.

## Data Availability

Not applicable.

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
