# Peer review of "Effect of Non-Thermal Food Processing Techniques on Selected Packaging Materials"

_polymers, 2022, doi:10.3390/polym14235069_

Round 1
Reviewer 1 Report
The work is a comprehensive collection of basic information on the non-thermal processing (NTP) techniques and selected packaging materials applicable in food industry. Descriptions of individual NTP methods as well as novel polymeric materials, are concise and in most cases the Authors highlight their essential advantages and disadvantages. The reviewed manuscript contains a large number of references to valid scientific works of the field.
In my opinion, a factor that hinders the reception of the paper, generally well-written and formulated, is the somewhat unclear way in which abbreviations are introduced and used in text. These acronyms refering to the names of individual NTP techniques, packaging materials or, finally, physical quantities can be found in such a numbers that render the text unclear (e.g., lines 134-139, 223-233). Also they can be found repeatedly used in a single sentence alongside proper names (e.g., lines 57-58, 155-156). In the subsequent paragraphs of the paper, the authors introduce more shortcuts, in my perception in a disorganized manner, forcing the reader to repeatedly check what a given acronym refers to. I would suggest, reducing the number of abbreviations used, e.g., by using them only in relation to NTP techniques and/or classes of polymeric materials, or at least distinguishing them consistently, e.g., by using uppercase letters for NTP techniques and lowercase letters for packaging materials.
In line 639, the authors write that UV radiation is a major cause of polymer degradation. This statement is true in regard to epoxies and chains containing aromatic functional groups. For example hydrocarbon-based polymers are susceptible to thermal degradation.
The authors are encouraged to adress above issues which hopefully could help improve the overall quality of the manuscript.
Author Response
Please see the attachement.

Reviewer 2 Report
Dear authors,
I consider your paper "Effect of non-thermal food processing techniques on selected packaging materials" can be published in Polymers after only a few minor revision.
1. There are some minor spell check required. For example:
…. inhomogeneity on the film surface of the and degradation - line 234
or
"respiration rat" - line 326
2. Maybe you can find a little bit more information on sections 3.1.4 - 3.1.6...?!
3. I think it will be better to introduce also some figures in the text.
